# Clinical Trials of Immune Checkpoint Inhibitors in Hepatocellular Carcinoma

**DOI:** 10.3390/jcm10122662

**Published:** 2021-06-16

**Authors:** Anne Dyhl-Polk, Marta Kramer Mikkelsen, Morten Ladekarl, Dorte Lisbet Nielsen

**Affiliations:** 1Department of Oncology, Herlev and Gentofte Hospital, University of Copenhagen, Borgmester Ib Juuls Vej 1, 2730 Herlev, Denmark; marta.kramer.mikkelsen@regionh.dk (M.K.M.); Dorte.Nielsen.01@regionh.dk (D.L.N.); 2Department of Oncology, Clinical Cancer Research Center, Aalborg University Hospital, Hobrovej 19-22, 9000 Aalborg, Denmark; morten.ladekarl@rn.dk; 3Faculty of Health and Medical Sciences, University of Copenhagen, Blegdamsvej 3B, 2200 Copenhagen, Denmark

**Keywords:** checkpoint inhibitor, immunotherapy, hepatocellular carcinoma, review

## Abstract

*Introduction*: Several immune checkpoint inhibitors (CPIs) are under clinical development in hepatocellular carcinoma (HCC) and the field is advancing rapidly. In this comprehensive review, we discuss published results and report on ongoing clinical trials. *Methods*: A literature search was carried out using PubMed and EMBASE; data reported at international meetings and clinicaltrials.gov were included as well. The search was updated 5 March 2021. We evaluated studies with monotherapy CPI’s, combinations of CPI’s and combinations of CPI’s with other treatment modalities separately. Only studies with at least 10 included patients were considered. *Results*: We identified 2649 records published in the English language literature. After review, 29 studies remained, including 12 studies with preliminary data only. The obtained overall response rate of PD-1/PDL-1 monotherapy in phase II studies in the second-line setting was 15–20% with disease control in approximately 60% of patients. The responses were of long duration in a subset of patients. Furthermore, the safety profiles were manageable. However, a phase III study comparing nivolumab with sorafenib in the first-line setting and a phase III study evaluating pembrolizumab versus best supportive care in the second-line setting did not meet their prespecified endpoints. More recently, a phase I/II study of nivolumab and ipilimumab has resulted in a response rate of approximately 30% with a median OS of 22 months in the second-line setting. Multiple trials have been initiated to evaluate CPIs in combination with molecularly targeted drugs, especially anti-angiogenic drugs or local therapy. A phase III study investigating atezolizumab plus bevacizumab versus sorafenib in the first-line setting showed significantly increased survival in the combination arm. *Conclusions*: The combination of atezolizumab and bevacizumab represents a new standard of care in the first-line setting for fit patients with preserved liver function. CPIs can produce durable tumor remission and induce long-standing anti-tumor immunity in a subgroup of patients with advanced HCC. Although phase III trials of CPI monotherapy have been negative, the combination of PD-1/PD-L1 inhibitors with other anti-angiogenic drugs, CTLA-4 inhibitors or other modalities may result in new treatment options for patients with HCC. Research on predictive biomarkers is crucial for further development of CPIs in HCC.

## 1. Introduction

Liver cancer is ranked as the fifth most common cancer in men and the ninth among women, with approximately 900,000 new cases worldwide in 2020, and liver cancer is the fourth leading cause of cancer-related deaths [1]. Hepatocellular carcinoma (HCC) accounts for 85–90% of all primary liver cancer cases [2,3]. HCC is often preceded by years of chronic inflammation with development of cirrhosis [4,5], mainly related to hepatitis B virus (HBV), hepatitis C virus (HCV) [3,6], alcohol or fatty liver disease [7,8,9].

Standard treatment for early stage HCC includes liver resection, liver transplantation and ablation (radiofrequency or microwave). Patients with intermediate stage disease are selected for trans-arterial chemo-embolization (TACE) and radio-embolization [10]. Although early diagnosis and treatment are important, most patients are diagnosed in advanced stage. Furthermore, patients receiving local therapy for early stage HCC are at high risk of recurrence [11]. First-line treatment for patients with advanced HCC and preserved liver function has consisted of the tyrosine kinase inhibitor (TKI) sorafenib, since the results of the SHARP [12] and Asian-Pacific (AP) [13] trials were published in 2008 and 2009, respectively. Sorafenib targets multiple kinases including vascular endothelial growth factor receptor (VEGFR), platelet-derived growth factor receptor (PDGFR) and RAF kinases. Although sorafenib has a positive effect on median overall survival (OS), the absolute benefit was marginal (2.8 months increase in the SHARP trial and 2.3 months increase in the AP trial) [12,13]. Since 2017, several other drugs primarily targeting the vascular endothelial system have become clinically available [14]. The TKI lenvatinib targeting VEGFR1, 2 and 3, as well as fibroblast growth factor receptors (FGFR) 1, 2, 3 and 4, PDGFR alpha, c-Kit, and the RET proto-oncogene was noninferior to sorafenib as first-line treatment [15], and in the second-line setting, the RESORCE phase III trial evaluating regorafenib, a multikinase inhibitor that inhibits the activity of VEGFR1, −2, −3, TIE2, PDGFR, FGFR, KIT, RET, RAF-1, and BRAF receptor tyrosine kinases, as well as the activity of Abl, has shown an OS of 10.6 months for patients treated with regorafenib versus 7.8 months for patients receiving placebo in patients who progressed on sorafenib [16]. More recently, cabozantinib targeting c-MET, VEGFR1, 2 and 3, and AXL has shown increased median OS compared to placebo (10.2 versus 8.0 months) in the second- or third-line setting [17]. Finally, a modestly increased median OS was demonstrated in the REACH-2 trial evaluating ramucirumab, a monoclonal antibody that inhibits ligand activation of VEGFR2, as second-line therapy for patients with advanced HCC and increased α-fetoprotein concentrations (median OS of 8.5 months versus 7.3 months in the intervention group compared to placebo) [18].

Recent advances in immune checkpoint inhibitors (CPIs) in other cancer types, e.g., melanoma and non-small cell lung cancer, have led to a paradigm shift in clinical practice highlighting their potential. Historically, patients with viral hepatitis and/or cirrhosis were excluded from clinical trials with CPIs due to fear of autoimmune hepatitis and viral reactivation. Promising early reports of the effect of CPIs targeting PD-1 in HCC with manageable toxicity paved the way for several large studies of CPIs alone or in combination with other drugs, including randomized clinical trials. Currently, several CPIs have either been approved or are under development. In this study, we provide an update of results of clinical studies with CPIs in HCC.

### 1.1. The Biology of Checkpoint Blockade

Under normal physiologic conditions, a plethora of suppressive pathways in the immune system exist to maintain inflammatory homeostasis, protect tissue integrity and protect against unwanted autoimmunity [19]. Tumors exploit these pathways, escaping immune detection and elimination [20]. The pathways are highly regulated by immune checkpoints. The two immune checkpoint receptors that have been most studied, cytotoxic T-lymphocyte-associated antigen 4 (CTLA4) and programmed cell death protein 1 (PD1), are both inhibitory receptors acting as brakes on CD3/CD28-dependent signaling. CTLA-4 primarily regulates the amplitude of the early stages of T cell activation. PD-1 signals through the PD-1 pathway, limiting T cell activation and effector T cell responses. CPIs in clinical use are antibodies against CTLA-4, PD-1 or its ligand PD-L1. The inhibitors modulate the interaction between tumor cells and cytotoxic T lymphocytes, whose function are thought to be exhausted [21]. Targeting CTLA-4 or PD-1/PD-L1 reverses the exhaustion of cytotoxic T lymphocytes, leading to elimination of tumor cells via re-induction of the “natural” function of the T cell population [22].

### 1.2. Hepatocellular Carcinoma as Target for Immunotherapy

From a clinical standpoint, there is evidence of an activated immune response in HCC, and by its nature, HCC is a natural target for immunotherapeutic approaches [23]: reduced infiltration of tumor infiltrating lymphocytes (TIL) in the tumor is correlated with neoplastic recurrence after liver transplantation [24]; presence of regulatory T cells (tregs) and cytotoxix T cells (CTL) in tumors has been correlated with patient survival. Thus, presence of low intratumoral Tregs and high intratumoral CTL has been shown to be a negative independent prognostic factor for overall survival (OS) [25]; different immune subtypes of the tumor microenvironment (TME) have been associated with histological and molecular classification of HCC [26]; and factors such as chronic inflammation [27] and cirrhosis maintain an immunosuppressive environment and induce T cell exhaustion [28]. Furthermore, immune evasion is a pivotal mechanism in the progression of the disease. It is beyond the scope of this review to summarize current data. The topic has been excellently reviewed by others [5,27,29].

## 2. Materials and Methods

This systematic review was prepared using the Preferred Reporting Items for Systematic Reviews and Meta-Analyses (PRISMA) guidelines. We performed a literature search in PubMed (1966 to 5 March 2021) and Embase (1974 to 5 March 2021). The search string is available in the Appendix A. All searches were restricted to publications in English. Finally, we searched clinical.trial.gov for ongoing studies.

Retrieved articles and abstracts were included if they met the following criteria: (1) patients with HCC, (2) prospective clinical trial, (3) treatment with a CPI and (4) data on efficacy. Ongoing studies with available interim data were included. We excluded case reports, letters to the editor, reviews, editorials, studies with ≤10 participants and studies where results could not be extracted separately for HCC.

The screening was performed by two authors (D.L.N. and A.D.-P.) independently. For relevant records, full text articles were obtained and reviewed for eligibility (D.L.N. and A.D.-P.). Disagreements in eligibility were discussed until consensus.

The following data were extracted: treatment, phase, number of patients, patient characteristics (performance status (PS), HCV and HBV infection, Child-Pugh score [30], Barcelona Clinic Liver Cancer (BCLC) stage [31,32], previous anticancer therapy), outcome including response rate (RR), duration of response (DOR), disease control rate (DCR), progression-free survival (PFS) or time to progression (TTP), OS and data concerning PD-L1 expression. Finally, we extracted overall adverse events (AEs), hepatic-related AEs, discontinuation rate and number of deaths due to toxicity.

## 3. Results

Our search identified 2649 records. Seventeen full-text articles and 12 conference abstracts met our inclusion criteria (Figure 1). A list of CPIs under evaluation in the treatment of HCC is given in Table 1 and results are summarized in Table 2. Overall toxicity and hepatotoxicity are given in Table 3.

### 3.1. Advanced/Metastatic Setting

#### 3.1.1. CTLA-4 Inhibition

##### Tremelimumab

Two studies investigated tremelimumab (Table 2). A phase II study [33] evaluated the drug (15 mg/kg every 90 days; current standard dose [34]) in 21 (17 evaluable) patients with HCC and chronic HCV infection. Totally, 24% of patients had received sorafenib and almost half of the patients had compromised liver function (Child-Pugh B (43%)). Partial response (PR) was registered in 18%, and 59% of the patients had stable disease (SD). In 45% of the patients, clinical benefit exceeded 6 months. Median TTP was 6.5 months, potentially biased by long tumor assessment intervals (evaluation every 90 days). Median OS was 8.2 months [33]. Notably, tremelimumab induced a significant decrease in viral load [35].

Tremelimumab at two dose levels (3.5 and 10 mg/kg every 4 weeks, up to six doses) followed by 3-month infusions (considered a suboptimal dose) [34] in combination with radiofrequency ablation, cryoablation or TACE was investigated in 32 patients (19 evaluable) with advanced HCC (Barcelona Clinical liver cancer (BCLC) C 78%) [36]. Sixty percent had intact liver function (Child-Pugh A), 69% had viral hepatitis and 66% of patients had received sorafenib. Totally, 26% of the patients experienced PR and 63% had SD, which in 45% lasted longer than 6 months. Median TTP and OS were 7.4 and 12.3 months, respectively [36].

##### Safety

No immune-related fulminant hepatitis occurred, and no drug discontinuation was necessary due to hepatitis. However, a transient increase in transaminases ≥ grade 3 was recorded in 45% of patients in the study by Sangro et al. [33], and in the study by Duffy et al. [36], 9 and 22% of patients experienced grade 3–4 elevations in aspartate aminotransferase (AST) and alanine aminotransferase (ALT), respectively.

##### Summary

The abovementioned studies showed an effect of CTLA-4 blockade with tremelimumab in HCC However, the limited sample sizes prevent any formal conclusion on the magnitude of this effect. The toxicity profile was manageable.

##### Ongoing Studies

We did not identify ongoing studies investigating monotherapy with CTLA-4 inhibitors.

#### 3.1.2. PD-1/PD-L1 Inhibition; Monotherapy

##### Nivolumab

The CheckMate 040 study [37] was a phase I/II trial of nivolumab that enrolled 262 patients (BCLC stage C 88% and Child-Pugh A 100%). The trial included patients with active HBV (24%; dose expansion phase) or HCV (23%; dose expansion phase) infections. Approximately three-fourths of patients had received one prior line of therapy, usually sorafenib.

In the dose-escalation phase, 48 patients were treated with nivolumab 0.1–10 mg/kg every 2 weeks in a 3 *+* 3 design. The RR was 15% with 6% complete responses (CR). Median DOR was 17.0 months and DCR 58%. Median TTP and median OS were 3.4 months and 15.0 months, respectively.

In the dose expansion phase, the standard dose of nivolumab (3 mg/kg on a 2-week cycle) was used. In this phase, 20% had a response and SD was reported in 45%. The DOR was particularly noteworthy as responses lasted for a median of 9.9 months. OS rates after 6 and 9 months were 83% and 74%, respectively. Median OS was only reached in the cohort of uninfected patients who had progressed on sorafenib and was assessed to be 13.2 months. Median PFS was 4.0 months. 

More recently, preliminary results from the Child-Pugh B cohort in the CheckMate 040 study were presented. Totally, 49 patients, of whom 51% had received prior sorafenib, received nivolumab. RR was approximate half of RR for Child-Pugh A (10%). However, the median DOR was comparable to that of Child-Pugh A patients (9.9 months). Median OS for sorafenib naïve and treated patients was 9.8 and 7.3 months, respectively [38].

In the phase III CheckMate 459 study, 743 patients with advanced HCC were randomized to nivolumab or sorafenib in the first-line setting. At a minimum follow-up of 22.8 months, the study did not meet the prespecified threshold of statistical significance (hazard ratio (HR) 0.84, *p* = 0.0419). Thus, median OS in the nivolumab arm was 16.4 months versus 14.7 months in the sorafenib arm (HR 0.85, 95% CI 0.72–1.02, *p* = 0.0752); the 12-month OS rates were 59.7% with nivolumab and 55.1% with sorafenib and the 24-month OS rates in the two groups were 36.8% and 33.1%, respectively. Median PFS was similar in the two arms (nivolumab: 3.7. months; sorafenib: 3.8 months), whereas RR was 15% versus 7% (*p*-values not reported). Patient-reported outcome suggested that patients receiving nivolumab experienced better quality of life, yet no details were reported [39].

##### Pembrolizumab

The KEYNOTE-224 phase II study investigated pembrolizumab in 104 patients with advanced HCC previously treated with sorafenib [40]. Patients had Child-Pugh A and approximately half of the patients were HBV (21%) or HCV (25%) positive. Response was recorded in 17%, and in 77%, the responses lasted for at least 9 months. Forty-four percent of the patients had SD. Median PFS was 4.9 months and median OS 12.9 months. Responses were similar in uninfected patients and patients with viral infections [41].

A phase II study of pembrolizumab in advanced HCC including 29 patients, approximately one-third of whom had previously received sorafenib, showed a RR of 32% with durable responses (>6 months) in 89% of responding patients. Median PFS was 4.5 months and median OS 13 months [42].

The KEYNOTE-240 phase III trial investigated pembrolizumab versus placebo plus best supportive care as second-line therapy [43]. Totally, 278 and 135 patients received pembrolizumab and placebo, respectively. The patients had Child-Pugh A and 80% had BCLC stage C. Totally, 26% and 16% were HBV and HCV positive, respectively. RR was 18.3% in the pembrolizumab arm compared to 4.4% in the placebo arm (*p* = 0.00007). DOR was durable (13.8 months) in the pembrolizumab arm. DCR was recorded in 62.2% of patients receiving pembrolizumab compared to 53.3% in the placebo arm. Median PFS was 3.0 months in the pembrolizumab arm compared to 2.8 months in the placebo arm; HR 0.718, *p* = 0.022 (pre-specified *p* = 0.002 required for statistical significance). Pembrolizumab was found to reduce the risk of death by 22%, and the efficacy was similar across subgroups. The median OS, however, was 13.9 months in the pembrolizumab arm compared to 10.6 months in the placebo arm; HR 0.78, *p* = 0.0238 (pre-specified *p* = 0.002 required for statistical significance). It has been suggested that subsequent anticancer therapy impacted the OS results. Thus, 41.7% of patients in the pembrolizumab arm and 47.4% (10.1% received a PD-1/PD-L1 inhibitor) in the placebo arm received subsequent therapy.

##### Camrelizumab

A phase II study of camrelizumab included 220 Chinese patients (98% Child-Pugh A) who were randomized to two different treatment schedules (3 mg/kg every second or third week). Totally, 83% of patients had HBV infection, and 97% had received previous treatment. The results were not significantly different between the two regimens. A RR of 14.7%, a PFS of 2.1 months and OS of 13.8 months were recorded [44].

##### Cemiplimab

The HCC expansion cohort of the phase I study evaluating cemiplimab included 26 patients who all except two had received at least one line of prior therapy. A PR was seen in 19.2% of patients and 53.8% had SD. Median PFS was 3.7 months [45].

##### Durvalumab

Durvalumab was evaluated in 40 patients with Child-Pugh A, 93% of which had received sorafenib. A RR of 10% was shown among all patients. Among HCV-positive patients, the RR was 25%, whereas no responders were recorded among 23 patients with HBV-positive disease. The DCR (SD ≥ 24 weeks) among all patients was 33.3%, 62.5% in HCV-positive patients and 11.1% in HBV-positive patients. The median OS for all patients was 13.3 months, for HCV positive 19.3 months and for HBV positive 6.3 months [46].

##### Avelumab

Avelumab was evaluated in a phase II study including 30 patients with Child-Pugh A, of which 87% had HBV infection and all had received prior sorafenib. PR was recorded in 10% of patients and DCR was 73%. Median TTP and OS were 4.4 and 14.2 months, respectively. Thus, the study did not meet its primary end point (RR of 15%) [47].

##### Safety

Except for a substantial increase in hepatic events, the safety profiles of nivolumab and pembrolizumab in patients with Child-Pugh A were comparable to that established for monotherapy in other tumor types [48] (Table 3).

In patients receiving nivolumab in the dose expansion phase of CheckMate 040, grade 3–4 AST or ALT increases were reported in 4% and 2%, respectively. Totally, 9% of patients had AEs leading to discontinuation [37]. In CheckMate 459, the number of cases with increased liver enzymes was not reported. However, grade 3–4 treatment-related adverse events (TRAE) were reported in 22% in the nivolumab arm, and 4% of patients discontinued treatment due to AEs [39]. For patients receiving pembrolizumab in KEYNOTE-224, increased AST or ALT grade 3–4 was reported in 4% and 2%, respectively, and immune-related hepatitis was seen in 3%. Five percent discontinued treatment due to an AE [40]. Additionally, increased AST or ALT grade 3–4 was reported in 13% and 6%, respectively, and hepatitis was seen in 3% of patients in the pembrolizumab arm in KEYNOTE-240 [43]. No cases of flares of HBV or HCV occurred in these studies. 

For patients with Child-Pugh B, 8% had hepatic TRAEs and 4% discontinued treatment due to TRAEs. Thus, in spite of a numerically higher number of hepatic events, the discontinuation rate was similar to what has been seen in patients with Child-Pugh A receiving nivolumab [37,38].

Camrelizumab and cemiplimab had safety profiles similar to other CPIs targeting PD-1 [44,45]. Except for a high incidence of immune-mediated reactive cutaneous capillary endothelial proliferation seen in patients receiving camrelizumab (67% compared to <3% with other PD-1 inhibitors) [44]. Although no grade 3–4 AEs were recorded this finding might be worrying. Finally, in patients receiving durvalumab or avelumab, no new safety issues were recorded, grade 3–4 elevated AST/ALT was reported in 13% of patients [46,47].

##### Summary

PD-1 inhibitors have demonstrated RRs of 15–30%, with durable responses and median OS of 13–16 months in the first- and second-line setting. However, phase III trials of nivolumab and pembrolizumab have failed to meet their primary endpoint (OS). PD-L1 inhibitors have only been evaluated in two phase II studies with disappointing RR of 10% and OS of 13–14 months. Except for a substantial increase in hepatic events (increase in liver enzymes), the safety profiles of PD-1/PD-L1 inhibitors were similar to that for monotherapy in other tumor types.

##### Ongoing Studies 

We identified 8 ongoing phase I, I/II or III trials evaluating monotherapy with PD-1/PD-L1 blockade in the advanced setting (Appendix A). The KEYNOTE-394 trial (NCT03062358) is almost identical to the KEYNOTE-240 trial. However, KEYNOTE-394 includes patients from Asia and will enroll an estimated 450 patients. Study completion is expected in June 2021. Also, the PD-1 inhibitor, tislezumab is investigated with sorafenib as compactor in a phase III study (RATIONALE-301; NCT03412773) using a non-inferiority design.

#### 3.1.3. PD-1/PD-L1 Inhibition in Combination with Antiangiogenic Agents

HCC is a highly vascularized tumor that exploits angiogenesis to grow and disseminate [49]. VEGF is one of the key players in angiogenesis and all approved targeted drugs in HCC inhibit VGEF(R) signaling. By targeting abnormal vessel formation antiangiogenic agents potentially increase infiltration of immune effectors cells. Thus, combination of CPI and VEGF(R) targeting therapies may be synergistic and scientific rational [50] (see discussion).

##### Pembrolizumab and Lenvatinib

A phase Ib trial of pembrolizumab plus lenvatinib including 100 patients in the expansion phase has shown a RR of 41% and a DCR of 86%in the first-line setting. The median DOR was 12.6 months, median PFS was 8.2 months and promising median OS of 22.0 months [51].

##### Camrelizumab and Apatinib

Apatinib is a new generation of small molecule anti-angiogenesis inhibitors that selectively binds to and inhibits VEGFR 2. In addition, this agent mildly inhibits c-Kit and c-SRC tyrosine kinases. A phase I study of camrelizumab plus apatinib, in patients who had previously received sorafenib showed a RR of 50% among 16 evaluable patients and a DCR of 94%. Median PFS was 5.8 months [52]. In addition, a phase II study in the first- and second-line setting evaluated the combination in 70 and 120 patients, respectively. In the first-line setting, a RR of 34% and a median PFS of 5.7 months were reported, whereas RR was 23% and PFS 5.5 months in the second-line setting. OS data were immature with 12-months survival rates of 75 and 68%, respectively [53].

##### Penpulimab and Anlotinib

Penpulimab is a novel humanized anti-PD-1 IgG1 antibody, which has been engineered to eliminate FC receptor binding activity in order to improve the efficacy [54]. This drug has been evaluated in patients with advanced HCC in combination with anlotinib, a multitargeted TKI selective for VEGF receptors 1/2/3, FGF receptors 1–4, PDGF receptors α and β, and c-kit. Totally, 31 patients of whom 77% had BCLC C were included in the first-line setting. Among 25 evaluable patients a RR of 24% and a DCR of 84% were reported. Median TTP was not reached, however, 6 months-TTP was 63% [55].

##### Atezolizumab and Bevacizumab

Recently, data from the HCC cohorts (group A and F) of the phase Ib GO30140 study of atezolizumab plus bevacizumab were published [56]. In group A all patients received atezolizumab plus bevacizumab. The group included 104 patients with Child-Pugh A, half of the patients had HBV infection and approximately one-third HCV infection. Totally, 36% of the patients had a confirmed objective response and 12% obtained CR. The responses were durable (23% lasting >6 months), and DCR was 71%, median PFS 7.3 months and median OS 17.1 months [56]. Group F of the same study included 119 patients with Child-Pugh A, half of the patients had HBV infection and approximately 20% HCV infection. The patients were randomized to atezolizumab plus bevacizumab vs atezolizumab monotherapy in the first-line setting. The two arms showed similar RRs (20 vs. 17%), but PFS was significantly increased in the combination arm (5.6 versus 3.4 months) (HR 0.55, *p* = 0.011). Thus, PFS of the combination in group F was shorter than that in group A (7.3 months). However, this finding may be explained by a shorter follow-up in the F group (median follow up 6.6 vs. 12.4 months). Survival data were immature as median OS was not reached in either treatment groups [56].

IMbrave150, a phase III study comparing azetolizumab plus bevacizumab with sorafenib in the first-line setting, randomized 501 patients with Child-Pugh A. Approximately half of patients had HBV infection and 20% HCV infection. The RR in the experimental arm was 29.8% with 7.7% CR vs. 11.3% with 0.6% CR in the sorafenib arm. SD was similar in the two arms (44.2 vs. 43.4%). A PFS of 6.8 months was observed in the atezolizumab plus bevacizumab arm vs. 4.3 months in the sorafenib arm (HR 0.59, 0.47–0.76, *p* < 0.0001), and median OS was 19.2 months vs 13.4 month (HR, 0.66, 95% CI 0.52–0.85, *p* = 0.0009) [57,58].

##### Avelumab and Axitinib

A phase Ib study investigated avelumab plus axitinib in treatment-naïve patients with HCC. Axitinib is a second generation TKI inhibiting VEGFR 1, 2, and 3, c-KIT and PDGFR. After inclusion of 22 patients, a RR of 14% and a SD of 68% were observed [59].

##### Durvalumab and Ramucirumab

The phase Ib study of durvalumab in combination with ramucirumab in patients with advanced HCC included 28 patients. A RR of 11% and a DCR of 61% were reported. Median DOR was not reached; however, the lower range was 5.6 months, indicating long-lasting responses. Median PFS and median OS were 4.4 and 10.7 months, respectively [60].

##### Safety

In general, it is reported that the safety profile of the combination therapy was similar to that reported for the monotherapies [57]. In the study of pembrolizumab plus lenvatinib, grade 3 and 4 TRAEs occurred in 67% of patients. Overall, 18% discontinued treatment due to TRAEs and 3% experienced grade 5 events [51]. In the GO30140 study, 68–82% of patients receiving atezolizumab and bevacizumab (group A and F) experienced any grade of toxicity and 20% of the patients (group F; TRAE not reported for group A) had grade 3 to 4 TRAEs, while 41% of patients receiving atezolizumab monotherapy experienced any grade TRAEs and 5% had grade ≥ 3 toxicity [56]. In Imbrave150, grade 3 and 4 TRAEs were reported in 56.5 and 55.1% of patients in the atezolizumab and bevacizumab arm and sorafenib arm, respectively. Hypertension was the most frequently reported TRAE with 15.2 and 12.2 grade 3 to 4 events in the combination and sorafenib arm, respectively. Approximately 15% in the atezolizumab and bevacizumab arm discontinued treatment due to AEs. Furthermore, 4.6% (15 patients) in the combination arm died due to side effects including three patients with gastrointestinal bleeding and two patients with abnormal liver function and liver injury, respectively [57]. Overall, the incidence of upper gastrointestinal bleeding in the atezolizumab and bevacizumab arm was 7% which is similar to other studies of bevacizumab in HCC [61]. The safety profile of the combination of camrelizumab and apatinib was reported to be similar to that reported for the monotherapies [52,53]. No AEs ≥ grade 3 were reported in the study of avelumab and axitinib, and no patients discontinued treatment due to AEs [59]. In the study of penpulimab plus anlotinib 10% of patients experienced grade 3 TRAE, it was reported that no unexpected AE were identified [55]. Also, in the study of durvalumab and ramucirumab, no unexpected toxicities were demonstrated [60]. In general, however, the regimens are associated with a high incidence of grade 3–4 toxicity (20 to 57%) with hypertension as the most common side effect counting for 5 to 50% of grade 3–4 events (Table 3).

##### Summary

In general, combinations of PD-1/PD-L1 inhibitors and VEGF(R) targeting therapy improved RR, PFS and OS compared with CPI monotherapy in the first-line setting. Most important, atezolizumab in combination with bevacizumab has shown significantly increased median OS compared to CPI monotherapy. The increased efficacy was obtained at the cost of a high incidence of grade 3–5 toxicity.

##### Ongoing Studies

Thirty-eight ongoing studies are investigating dual CPI and antiangiogenic agents, seven of which are phase III randomized studies. The phase III study (LEAP-002; NCT03713593) of lenvatinib *+*/− pembrolizumab will provide data on the efficacy of combination therapy. Results are expected in July 2022. A phase III study (NCT03764293) compares camrelizumab in combination with apatinib with sorafenib monotherapy. Two phase III studies (COSMIC-312; NCT03755791 and IMbrave251; NCT04770896) evaluate atezolizumab in combination with different VEGFR inhibitors (lenvatinib, sorafenib and carbozantinib). Furthermore, sintilimab in combination with the VEGF-antibody, IBI305 is compared to sorafenib (NCT03794440). Finally, toripalimab is investigated in two studies in combination with lenvatinib and bevacizumab, respectively (NCT04523493; NCT04723004).

#### 3.1.4. CTLA-4 and PD-1/PD-L1 Inhibition

The combination of CLTLA-4 and PD-1/PD-L1 inhibition has been found to produce a higher incidence of durable responses across several malignancies, albeit at the expense of a dose dependent increased rates of immune-related AEs [62].

##### Nivolumab and Ipilimumab

Recently, results for the combination of nivolumab and ipilimumab in the CheckMate 040 trial (three different dosing regimens) were reported. Totally, 148 patients who had previously received sorafenib were included. More than 90% of patients had BCLC stage C. The RR, DOR and DCR were similar across treatment arms. Overall OR was 31% with 5% CR. DCR was reported to be 49%. Nivolumab 1 mg/kg *+* ipilimumab 3 mg/kg Q3W (4 doses) followed by nivolumab 240 mg Q2W was found to have a promising median OS of 22.8 months, whereas the median OS in the other cohorts was 12.5 and 12.7 months, respectively [63].

##### Durvalumab and Tremelimumab

A phase I/II study of durvalumab and tremelimumab included 40 patients, of whom 50% were HBV or HCV positive and 30% had no prior systemic therapy [64]. The RR was 15% in all patients and 30% in uninfected patients. No patients with viral infection had response. The DCR (SD ≥ 16 weeks) was 58% in all patients and 70% in uninfected patients. Preliminary results from the expansion cohort of the same study included 332 patients treated in the first-line setting. The patients were randomized to four arms: A single dose of tremelimumab plus durvalumab, tremelimumab plus durvalumab, and monotherapy of durvalumab and tremelimumab, respectively. The RR were 22.7, 9.5, 9.6 and 7.2% with median OS of 18.7, 11.3, 11.7, and 17.1 months. Thus, highest clinical activity was found for the single dose tremelimumab (priming dose) plus durvalumab regimen [65].

A phase II study of the combination in patients with HCC or biliary cancer reported preliminary data on 10 patients with HCC. A RR of 50%, a PFS of 7.8 months and a median OS of 15.9 months were reported [66].

##### Safety

The combination of nivolumab and ipilimumab resulted in 31–53% 3–4 grade TRAEs, and 2–18% of patients discontinued treatment due to toxicity [63]. Patients receiving nivolumab 1 mg/kg plus ipilimumab 3 mg/kg Q3W followed by nivolumab 240 mg Q2W experienced the highest rates of adverse events, immune-mediated events and discontinuation rate due to toxicity. Totally 20% of patients receiving this combination developed hepatitis. On the other hand, the toxicity was manageable and 90% of hepatic events resolved using protocol-specified management algorithms. 

In the phase I/II study of durvalumab and tremelimumab, the most common grade ≥ 3 related AE was asymptomatic increased AST (10%). In all, 7.5% of patients discontinued treatment due to AEs [64].

##### Summary

The best performing combination regimen of nivolumab and ipilimumab resulted in increased RR of 31% with a median OS of 22 months in the second-line setting whereas the best performing regimen of durvalumab plus tremelimumab showed a RR of 23% and a median OS of 19 months in the first-line setting. Except for hepatotoxicity, the safety profile of the combinations was consistent to that found in studies investigating the combination in other tumor types [67].

##### Ongoing Studies

Six ongoing studies are investigating the CPI combination strategy. The phase III CheckMate 9DW (NCT04039607) is scheduled to include 1084 patients randomized to nivolumab and ipilimumab vs sorafenib or lenvatinib. Results are expected in September 2023. Additionally, sintilimab in combination with the CTLA-4 inhibitor, IBI310 is compared to sorafenib in a phase III study expected to include 490 patients (NCT04401813). Finally, a phase III study, HIMALAYA, will compare the efficacy of durvalumab plus single dose tremelimumab with durvalumab monotherapy vs sorafenib (NCT03298451).

#### 3.1.5. PD-1/PD-L1 Inhibitor in Combination with Chemotherapy

Preliminary data from a phase II study of camrelizumab in combination with FOLFOX4 (5-fluorouracil, leucovorin, oxaliplatin) or GEMOX (gemcitabine, oxaliplatin) in patients with advanced HCC or biliary tract cancer showed a RR of 27% among 27 patients with HCC and a DCR of 79%. The median PFS was 5.5 months. Notably, 85% of patients experienced grade 3–4 toxicity [68].

#### 3.1.6. PD-1/PD-L1 Inhibitor in Combination with Local Therapy

##### Efficacy

A proof-of-concept study enrolled 50 patients with advanced HCC (74 extrahepatic metastases), who had previously received or had unacceptable toxicity of sorafenib [69]. Thirty-three of the patients with stable disease or mixed response to single agent PD-1 inhibitor (nivolumab or pembrolizumab) received subtotal thermal ablation. Among 50 patients treated with PD-1 inhibitor 10% of patients had a response and 42% SD. Additional ablation increased RR to 24%. Median PFS for all patients was 5 months and OS 16.9 months [69]. The combination of Y90-radioembolization and nivolumab has been investigated in a study of 40 patients with advanced HCC of which 64% had BCLC C and 14% had received prior systemic therapy. The study showed an encouraging RR of 31%. PFS and OS were 4.6 months and 15.1 months, respectively [70].

##### Safety

The combination of CPI with local therapy seems well tolerated, with no new safety issues [69,70].

##### Summary

Two small studies indicate that local therapy might increase the efficacy of CPI, however, the limited sample sizes of the studies prevent any formal conclusion on the efficacy of this combination.

##### Ongoing Studies

A range of phase I and phase II trials are investigating different CPIs in combination with other agents or other treatment strategies, but we identified only two phase III studies. In a phase III study, camrelizumab plus FOLFOX is compared to FOLFOX or sorafenib (NCT03605706). Furthermore, a phase III study is planned to compare toripalimab in combination with radiotherapy to sorafenib in patients with HCC, BCLC stage C and portal vein thrombosis (NCT04709380).

### 3.2. Locally Advanced Setting

No results were reported. Twenty-five studies including three phase III studies are ongoing in this setting combining CPI with Y90 microspheres (SIRT), ablation or transarterial chemoembolization (TACE) (Appendix A). Most interesting, a placebo-controlled phase III study expected to include 765 patients compares nivolumab with and without ipilimumab in combination with TACE to TACE alone (NCT04340193). Additionally, a placebo-controlled phase III study is investigating chemoembolization in combination with durvalumab or durvalumab plus bevacizumab (EMERALD-1; NCT03778957). Finally, a phase III study is evaluating atezolizumab plus bevacizumab in combination with TACE with TACE alone (NCT04712643).

### 3.3. Preoperative and Adjuvant Setting

#### 3.3.1. Efficacy and Safety

The phase II randomized study of nivolumab vs nivolumab plus ipilimumab as preoperative therapy in 27 patients with resectable HCC reported a pathological complete response (pCR) rate of 24% among 21 patients who proceeded to surgery (surgery was aborted in 6 patients including 3 patients with progressive disease). Furthermore, 16% had major pathological responses. Among all randomized patients the pCR rate was 19% (preliminary data). Totally, 5% and 24% of the patients experienced grade 3 toxicity. Surgery was not delayed or cancelled due to toxicity [71].

#### 3.3.2. Ongoing Studies

We identified ten ongoing studies including two ongoing randomized phase II studies (NCT03510871; NCT03222076) in the neoadjuvant setting. In the adjuvant setting, thirteen trials were identified (two were also neoadjuvant) including seven phase III trials (Appendix A).

Importantly, three phase III studies evaluating monotherapy with nivolumab (CheckMate 9DX; NCT03383458), pembrolizumab (KEYNOTE-937; NCT03867084), and toripalimab (NCT03859128), respectively, as adjuvant therapy after curative resection or ablation are in progress. The studies will each include 530, 950 and 402 patients and are expected to be completed in 2022, 2025, and 2022 respectively. Three studies are investigating the combination of CPI and VEGF(R) inhibitor: NCT04682210 (sintilimab *+* bevacizumab vs. active surveillance), NCT04639180 (camrelizumab plus apatinib vs. active surveillance) and NCT04102098 (IMBrave 050; atezolizumab in combination with bevacizumab vs. active surveillance). Each study is planned to include 246, 674 and 662 patients and to be completed in 2023, 2024 and 2023, respectively. Furthermore, a fourth study investigating durvalumab alone and in combination with bevacizumab (EMERALD-2) in the same setting is underway. This study is expected to include 888 patients and to be completed in June 2022.

### 3.4. Biomarkers

PD-L1 expression level was investigated in eight studies with divergent results (Table 4) [37,39,40,41,42,44,52,60]. In three studies, only 14–50% of the samples were available [40,42,44], and one study included tumor types other than HCC [52]. Furthermore, in most studies no statistical analyses were performed. In CheckMate 40, 81% of the patients in the dose-expansion phase were investigated. PD-L1 ≥ 1% on tumor cells was expressed in 20% of assessed tumors and in 26% (95% confidence interval (CI) 13–44%) of tumors with objective response. No correlation was reported [37]. In contrast, evaluation of PD-L1 in CheckMate 459 showed a higher RR in PD-L1-positive patients receiving nivolumab (PD-L1 ≥ 1% RR 20/71 (28%); PD-L1 < 1% RR 36/295 (12%)) (no p-values were reported) [39].

In Keynote-224, a retrospective analysis of PD-L1 expression with available data (half of the patients) revealed that PD-L1 expression assessed by a combined positive score (CPS) (a measure of PD-L1 positive immune and tumor cell number) was associated with response. In contrast, the tumor positive score (TPS) did not correlate with response [40,41].

In the study of camrelizumab, PD-L1 expression data were only available in 30 of 220 patients. The RR was reported in 36% of 11 patients with PD-L1 ≥ 1% and in 11% of 19 patients with PD-L1 < 1% [44]. No responses were reported among 15 patients with low expression of PD-L1 on circulating tumor cells in the study of camrelizumab and apatinib [52]. In contrast, objective responses were observed irrespectively of PD-L1 expression in the GO30140 study. Furthermore, patients with PD-L1 positive tumors seem to have the shortest PFS. However, evaluating the association between PD-L1 and efficacy was challenging due to small numbers of patients in some of the subgroups [56]. Finally, response rates and median OS were comparable regardless of PD-L1 expression in the nivolumab plus ipilimumab part of CheckMate 040 [63]. In the study of camrelizumab and apatinib [52], patients with PR/SD at first response evaluation showed significantly higher TMB than those with PD (mean 8.53 vs. 1.44 mutations/MB; *p* = 0.0002); however, PFS was not significantly different (*p* = 0.063).

## 4. Discussion

The promising results of early studies with PD-1 monotherapy in HCC generated much enthusiasm [37,40]. In 2017 and 2018 FDA granted accelerated approval to nivolumab and pembrolizumab, respectively, for second-line therapy in advanced HCC. Nonetheless, the KEYNOTE-240 trial investigating pembrolizumab vs placebo plus best supportive care in the second-line setting did not meet its coprimary endpoints [43]. Furthermore, the phase III CheckMate-459 head-to-head trial of nivolumab vs sorafenib in the first-line setting did not meet its pre-specified primary outcome (OS) [39]. Anyhow, a high frequency of subsequent use of systemic therapy including immunotherapy in the sorafenib arm might have blurred the results. Recently (11 March 2021), FDA announced that the agency would discuss (27–29 April 2021) indications granted with accelerated approval for pembrolizumab, atezolizumab, and nivolumab [72].

Whereas, nivolumab and pembrolizumab have partly overlapping binding epitopes camrelizumab has been found to interact with a unique epitope of the PD-1 molecule [73]. However, results regarding monotherapy with camrelizumab are similar to those of nivolumab and pembrolizumab.

Taken together, current results on CPI monotherapy are unlikely to change the standard of care in treatment of HCC, primarily due to lack of reliable predictive factors. Although all studies have shown relatively high response rates and durable responses, at least 30% of patients have unequivocal progression on CPI monotherapy and all patients progress at some point. Therefore, development strategies to improve outcomes have become urgent. The strategies have frequently been based on drug combinations that include drugs targeting the vascular endothelial system, including drugs already proven efficient in HCC (sorafenib, lenvatinib, regorafenib, ramucirumab and cabozantinib).

Several factors constitute the rationale for combination of CPIs targeting PD-1/PD-L1 and antiangiogenic agents: VEGF(R) inhibition promotes maturation of dendritic cells and modulates multiple effectors including cytotoxic and regulatory T cells and natural killer cells resulting in an effective priming and activation of T cells [74,75]; the combination anti-PD-1/PD-1 and antiangiogenic therapy normalizes vessel formation and promotes infiltration of T cells into the tumor [76,77]; antiangiogenic therapy inhibits the activity of immunosuppressive cells (regulatory T cells (tregs), tumor associated macrophages (TAMs) and myeloid-derived suppressor cells (MDSCs)) changing an immunosuppressive microenvironment to an immunostimulatory environment [78]; and anti-PD1/PD-L1 therapy enhances the ability of T cells to attack tumor cells [76,77].

Several studies have shown impressive response rates, durable responses and favorable PFS using the combination of a CPI targeting PD-1/PD-L1 and VEGF(R) inhibition. Recently, data from the HCC cohorts (group A and F) of the phase Ib GO30140 study of atezolizumab plus bevacizumab were published [56]. In group F patients were randomized to atezolizumab plus bevacizumab vs atezolizumab monotherapy in the first-line setting. The two arms showed similar RR, however, PFS was significantly increased in the combination arm (5.6 vs. 3.4 months). Furthermore, results from the IMbrave150 study comparing the combination of azetolizumab and bevacizumab with sorafenib in the first-line setting have shown significant improvements in PFS and median OS in the experimental arm. The combination was approved of FDA in 2020 and results have been groundbreaking changing standard of care in the first-line setting. Although, the toxicity reported was not trivial, with 57% grade 3 or 4 AEs [57], guidelines from e.g., ASCO recommend atezolizumab plus bevacizumab to be offered to most patients in the first-line setting [79].

In addition, a phase Ib trial of pembrolizumab plus lenvatinib have showed a RR of 41% in the first-line setting. Median PFS was 8.2 months and OS 22.0 months [51]. Although clinically meaningful efficacy was achieved in this study, the combination is not approved by the US Food and Drug Administration (FDA) as the “application request did not show sufficient evidence that the pembrolizumab plus lenvatinib combination represented a “meaningful advantage” over already available therapies” [80].

Although results concerning combination therapy with anti-PD-L1/PD1 inhibitors and angiogenenic therapy have been impressive, some crucial issues persist, such as validation in Western patients, because they present a different molecular profile of HCC (75% of patients in GO3014 were Asian and 40% of patients were Asian (excluding Japan)). Another consideration may be safety in unselected patients. Hence, the effect of TKIs may be dose and agent dependent [81]. Preclinical studies have shown that lower doses of TKIs were superior to higher doses in inducing homogeneous tumor vessel normalization [82]. The use of lower, vascular-normalizing doses of anti-VEGF therapies is supported by emerging clinical data in other tumor types [83] but needs further confirmation in studies in HCC.

The PD-1/PD-L1 and CTLA-4 pathways have distinct but complementary roles in negatively regulating immune activity (see “Section 1.1”). Thus, it seems biologically rational to combine inhibitors of the two pathways. Nivolumab plus ipilimumab has proved effective in the treatment of other tumor types [84,85]. Few studies have investigated the combination of a PD-1/PD-L1 inhibitor and a CTLA4 inhibitor in HCC. Most important, the combination of nivolumab and ipilimumab in the CheckMate 040 trial provided a robust clinical benefit [63]. Patients in the combination arm with the highest dose of ipilimumab had the highest CR rate and the most promising median overall survival (22.8 months). These results suggest that an increased dose of ipilimumab may translate into higher rates of durable responses and improved survival in patients with HCC. Similar results have been reported for other tumor types e.g., melanoma, renal cell carcinoma [86,87]. This combination was approved by FDA for HCC in 2020.

More recently, a systematic review including eight studies (6290 patients) in the first-line setting and six (2653 patients) in the second-line has been published. In the first-line setting, network meta-analysis showed the combination of atezolizumab and bevacizumab was superior compared with lenvatinib (HR 0.63, 95% CI 0.44–0.89), sorafenib (HR 0.58, 95% CI 0.42–0.80), and nivolumab (HR 0.68, 95% CI 0.48–0.98) with regard to OS. In the second-line setting, the analyses showed that all studied drugs had a PFS benefit compared with placebo. However, this only translated into OS benefit with regorafenib, cabozantinib and ramucirumab compared with placebo [88].

A different approach to improve the response is to modulate the immunogenicity of tumors or to boost the immune system by combination of locoregional and/or radiotherapy with immunotherapy. Thus, local therapy, such as radiofrequency ablation, TACE, SIRT or radiotherapy, may promote an immune response via the influence of the TME [89,90,91]. This approach is based on releasing tumor antigens through cell death induced by locoregional therapy, which subsequently improves immunotherapy due to better antigen presentation. A single, small, unrandomized study evaluating subtotal thermal ablation in combination with PD-1 inhibition in patients with advanced disease has shown an increased response rate after ablation supporting this treatment strategy [69]. This leaves a hypothetical role for CPI given in a combination with local treatment. Several open questions remain to be answered regarding patient selection, optimal timing and sequence of therapies and choice of combination. It will take several years before mature survival data become available [92]. Furthermore, translational studies are also needed to improve the understanding of the exact molecular mechanisms involved in the response or failure of these combinations.

We identified a huge number of ongoing studies, but only a few randomized, potentially practice-changing studies were identified. The scientific rationale behind parallel testing of multiple drugs in different combinations may be questioned. There is a need for rational/coordinated development of immunotherapy across the heterogeneous treatment landscape. On the other hand, the many generic drugs developed for the same indication will definitely increase competition and probably result in lower prices.

### 4.1. Biomarkers

Among the most investigated predictive biomarkers for CPI blockade are PD-L1, microsatellite instability/defective mismatch repair (MSI/dMMR) and tumor mutational burden (TMB). CPI is approved for clinical use in tumors with MSI/dMMR irrespective of tumor type, however, in two large series (122 patients and 82), the incidence of MSI in HCC was very low (0 and 2.4%) [93,94]. TMB representing genomic instability is emerging as a potential predictor of response to immunotherapy [95,96]. In general, only a few HCC tumors are TMB high [97]. TMB has only been reported in one study [52]. Therefore, the value in HCC is unclear.

PD-L1 expression on tumor cells predicts efficacy of PD-1/PD-L1 inhibitors in several tumor types. More recently, guidelines for implementation and interpretation of PD-L1 expression has been published [98]. However, the value of PD-L1 as the “definitive” biomarker is controversial [99,100]. Regarding testing, many unsolved issues exist including the use of different staining platforms and antibodies, the type of cells in which PD-L1 is assessed and thresholds for PD-L1-positivity.

Limited data on PD-L1 expression in HCC have been reported. Not surprisingly, hepatocytes and Kupffer cells have been found to express high levels of PD-L1 [101,102,103,104]. However, a significant inter-assay discordance in the quantitation of PD-L1 level in HCC has been reported [105]. Additionally, a significant heterogeneity was found in PD-L1 expression in the KEYNOTE-224 study [40,41]. In all reported studies, clinically meaningful responses were observed in patients with PD-L1 expression < 1% (Table 4). Anyhow, patients with a high PD-L1 expression had a numerically increased response rate in the CheckMate 040 study and PD-L1 expression was associated with improved OS (median 28.1 vs. 16.6 months) [106]. In addition, patients with a high PD-L1 expression had a numerically increased response rate in the CheckMate 459 study, suggesting that the marker might be reliable in HCC [39]. Taken together, there might be an association of PD-L1 expression with the efficacy of anti-PD-1/PD-L1 immunotherapy in HCC and PD-L1 expression might be a valuable predictor of the efficacy of anti-PD-1/PD-L1 therapy in certain patients, however, analyses from especially combination studies showed that patients benefited from immunotherapy regardless of PD-L1 expression level.

The immune response, however, is the result of multiple factors including the antigenic characteristics of the tumor and the multiplicity and phenotypes of tumor-associated antigen-specific cytotoxic T lymphocytes. From this perspective, the use of a single biomarker seems insufficient [107]. Several other candidate biomarkers have been suggested including tumor infiltrating lymphocytes, CD8 *+* T cell density in the TME, IFN_γ_ signaling, Wnt/β-catenin signaling and genetic signatures [108,109]. More recently, Sangro et al. found an inflammatory gene signature consisting of four genes to be associated with RR (*p* = 0.05) and OS (*p* = 0.01) in CheckMate 040 (37 of 262 patients included in analysis). The findings have not been validated [106].

Even though several serum biomarkers have been suggested and liquid biopsy is emerging as a clinical tool [110,111], there are currently no validated predictive biomarkers able to guide treatment choice and identification of predictors of response represents a major challenge [112]. Furthermore, the routine clinical practice of largely abandoning histological confirmation of the diagnosis in favor of radiologic criteria [113] is in conflict with a future perspective integrating biomarker information for therapeutic stratification of patients with HCC.

### 4.2. Patient Selection

With a few exceptions CPI is or has been investigated in patients with HCC Child-Pugh A. Preliminary results from the Child-Pugh B (7 *+* 8) cohort of the CheckMate 040 study investigating nivolumab showed an RR (10%) of approximate half of RR for Child-Pugh A patients. Furthermore, an increased number of hepatic adverse events were demonstrated in the Child-Pugh B cohort [38]. A retrospective study including 132 and 71 patients classified as Child-Pugh A and B, respectively, found a lower RR (2.8% vs. 15.9%, *p* = 0.010) and shorter OS (11.3 weeks vs 42.9 weeks, *p* < 0.020) for patients with Child-Pugh B compared to patients with Child-Pugh A [114]. Additionally, a systematic review of 597 patients with HCC found a significant correlation between number of grade ≥3 AEs and the proportion of patients with Child-Pugh B [115]. Although toxicity was manageable and the discontinuation rate was similar for Child Pugh A and B patients in ChechMate 040, these findings indicate that CPI should not be used in unselected patients with Child-Pugh stage B.

### 4.3. Safety

Although cirrhosis was present in >80% of the patients, the early fear of immune-mediated hepatitis did not pan out. Brown et al. performed a literature review on monotherapy ipilimumab, tremelimumab, nivolumab or pembrolizumab for patients with HCC, melanoma and non-SCLC [48]. Although, the study showed a substantial increase in hepatic events in patients with HCC as compared to patients with melanoma or non-SCLC, there was no significant difference among the groups with respect to treatment discontinuation or deaths secondary to drug toxicity. Another concern has been hepatitis B and C reactivation. A systematic review found a very low degree of virus activation. However, the authors recommend patients with active viral hepatitis to be monitored closely and treated with antiviral therapy if indicated [116].

Patients receiving a combination CPI and VEGF(R) inhibitor experienced a high number of grade 3–4 TRAE (Table 3). However, in the IMbrave150 study AE grade 3 to 5 TRAEs tended to occur at same rate in the sorafenib arm and the atezolizumab plus bevacizumab arm [57]. In addition, findings from a systematic review of pembrolizumab plus lenvatinib vs. pembrolizumab and lenvatinib monotherapies in a range of different cancer types has indicated that the combination did not increase drug toxicities and that the toxicity profile was manageable [117].

The safety profile of combination therapy with nivolumab and ipilimumab was consistent with studies investigating the combination in other tumor types [67]. Anyhow, ASCO guidelines underscore the risk of life-threatening toxicity [79].

Finally, a systematic review and a pooled analysis of 2403 patients with unresectable HCC receiving CPIs concluded that CPIs are safe. However, among 15 patients who received CPI in the setting of liver transplant, fatal graft rejection was reported in 40% and the mortality rate was 80%. Thus, the authors warranted caution regarding use of CPI in this setting [118]. 

## 5. Conclusions

The combination of atezolizumab and bevacizumab has resulted in a paradigm shift in the treatment of HCC and has changed practice in first-line setting. CPIs have the potential to produce durable tumor remission and induce long-standing anti-tumor immunity in a subgroup of patients with HCC. Results of CPIs in localized HCC in combination with local therapies and as neoadjuvant or adjuvant therapy in resectable disease are awaited. Research for predictive biomarkers is crucial for further development of this treatment modality.

**Table 1 jcm-10-02662-t001:** Immune checkpoint inhibitors under evaluation in clinical trials in hepatocellular carcinoma.

Compound	Company	IgG Class
**CTLA-4 inhibitor**		
Ipilimumab, Yervoy^®^ (BMS-734016)	Bristol-Meyers Squibb (New York, NY, USA)	IgG1, fully human
Tremelimumab (MEDI1123, formerly known as ticilimumab)	MedImmune (Gaithersburg, MD, USA)/ AstraZeneca (Cambridge, UK)	MedImmune, AstraZeneca
**PD-1 inhibitor**		
Nivolumab, Opdivo^®^ (BMS-936558, MDX-1106)	Bristol-Meyers Squibb (New York, NY, USA)	IgG4, fully human
Pembrolizumab, Keytruda^®^ (MK-3475, lambrolizumab)	Merck (MSD) (Kenilworth, NJ, USA)	IgG4, humanized
Tislelizumab (BGB-A317)	BeiGene Boehringer Ingelheim	IgG4, humanized
Camrelizumab (SHR-1210)	Jiangsu HengRui (Lianyungang, China)/Incyte (Wilmington, DE, USA)	IgG4, humanized
Cemiplimab, Libtayo^®^ (REGN2810)	Regeneron (Tarrytown, NY, USA)/Sanofi Genzyme (Cambridge, MA, USA)	IgG4, fully human
Spartalizumab (PDR001)	Novartis (Basel, Switzerland)	IgG4, humanized
Sintilimab, Tyvyt^®^ (IBI308)	Innovent Biologics (Suzhou, China)/Eli Lilly (Indianapolis, IN, USA)	IgG4, fully human
Toripalimab (JS001)	Shanghai Junshi Biosciences (Shanghai, China)	IgG4, humanized
Penpulimab (AK105)	Akeso Biopharma (Zhongshan, Anhui, China)	IgG1, humanized
**PD-L1 inhibitor**		
Atezolizumab, Tecentriq^®^, MPDL3280A	Genentech (South San Fransico, CA, USA)/Roche (Basel, Switzerland)	IgG1, fully humanized
Durvalumab, Imfinzi^®^, (MEDI4736)	MedImmune (Gaithersburg, MD, USA)/AstraZeneca (Cambridge, UK)	IgG1, fully human
Avelumab, Bavencio^®^, (MSB0010718C)	Merck Serono (Darmstadt, Germany)/Pfizer (New York, NY, USA)	IgG1, fully human
Lodapolimab (LY3300054)	Eli Lilly (Indianapolis, IN, USA)	IgG1, fully human

**Table 2 jcm-10-02662-t002:** Efficacy of checkpoint inhibitors in hepatocellular carcinoma.

	Treatment	Phase	Number of Patients (Evaluable)	Patient Characteristics	Previous Systemic Therapy (%)	Response Rate (%) (95% CI)	Median PFS/TTP (Months) (95% CI)	Median OS (Months) (95% CI)
**Advanced, not amenable for resection or ablation**								
**CTLA-4 inhibition**								
Sangro [33]	Tremelimumab	II	21 (17)	PS	Sorafenib 24 Any prior treatment 57	Confirmed PR 17.6 SD (≥12 weeks) 58.8 DCR 76.4	PFS 6.5 (4.0–9.1)	8.2 (4.6–21.3)
				0 71 1 29				
				HCV+ 100				
				Child-Pugh				
				A 57 B 43				
				BCLC stage A 14 B 29 C 57				
				Extrahepatic disease 10				
Duffy [36]	Tremelimumab *+* RFA/CA/TACE	Pilot study	32 (19)	PS	Sorafenib 66	Confirmed PR, (hepatic disease only; n = 19)	TTP 7.4 (4.7–19.4) (n = 28)	12.3 (9.3–15.4) (n = 28)
				0 88 1 13				
				HBV+ 16 HCV+ 59		26 (9.1–51.2)		
				Child-Pugh 5 44 6 16 7 9				
				NR 31	Other systemic therapies 28	SD 63		
				BCLC stage B 22 C 66				
				NR 12		DCR 89		
				Extrahepatic disease 44				
**PD-1/PD-L1 inhibition; monotherapy**								
El-Khouiry [37] (CheckMate 040)	Nivolumab	I	48	PS	Sorafenib 74 Other systemic therapy 9	RR 15 (6–28) (CR 6)	TTP 3.4 (1.6–6.9)	15.0 (9.6–20.2)
				0 60				
				1 40				
				HBV+ 31				
				HCV+ 21				
				Child-Pugh 5 85 6 15				
				Extrahepatic disease 71				
	Nivolumab	II	214	PS	Sorafenib 68	RR 20 (15–26) (CR 1) DCR 64 (58–71) DOR 9.9 months (8.3-NE)	4.0 (2.9–5.4)	NR 6-month OS 83% (78–88%) 9-month OS 74% (67–79%)
				0 64				
				1 36				
				HBV+ 24				
				HCV+ 23	Other systemic therapies 6			
				Child-Pugh				
				5 70 6 29 > 6 2				
				Extrahepatic disease 53				
Kudo [38]	Nivolumab	II (preliminary data)	49	PS 0–1	Sorafenib 51	RR 10.2	NR	7.6
(CheckMate 040; Child-Pugh B cohort)				Child-Pugh B7–8		DCR 55.1		
				Vascular invasion or extrahepatic disease 57.1		DOR 9.9 months		
Yau [39] (CheckMate 459)	Nivolumab	III (preliminary data)	371 vs. 372	Not eligible for surgical or locoregional therapies	No prior systemic therapy	RR 15 (CR 4) vs.	3.7 (3.1–3.9) vs. 3.8 (3.7–4.5)	16.4 (13.9–18.4) vs. 14.7 (11.9–17.2)
						7 (CR 1)		
	Sorafenib			PS 0–1		SD 35 vs. 48		HR 0.85 (0.72–1.02); *p* = 0.0752
				Child-Pugh A		DOR 23.3 months (3.1–34.5 *+)* vs. 23.4 months (1.9–28.7 *+)*		
Zhu [40] (Keynote-224)	Pembrolizumab	II	104	PS	Sorafenib 100	RR 17 (11–26) (CR 1) SD (≥6 weeks) 44 DCR 62 (52–71) DOR 2.1 months (2.1–4.1) DOR ≥ 9 months 77	PFS 4.9 (3.4–7.2)	12.9 (9.7–15.5)
				0 61				
				1 39				
				HBV+ 21				
				HCV+ 25				
				Child-Pugh				
				A 94				
				B 6				
				BCLC stage				
				B 24%				
				C 76%				
				Extrahepatic disease 64				
Feun [42]	Pembrolizumab	II	29 (evaluable 28)	PS	Sorafenib 34	RR 32 (15.9–52.4)	PFS 4.5 (2–7)	13 (7- NE)
				0 52				
				1 48				
				HBV+ 17		SD 14		
				HCV+ 31				
				Child-Pugh				
				A 97		DCR 46		
				B 3				
				Extrahepatic disease 72				
Finn [43] (Keynote-240)	Pembrolizumab vs. Best supportive care	III	278 vs. 135	PS	Sorafenib 100	RR 18.3 (14.0–23.4) vs. 4.4% (1.6–9.4) *p* = 0.00007 DCR 62.2 vs. 53.3 DOR 13.8 months (1.5–23.6 *+)* vs. Not reached (2.8–20.4 *+)*	PFS 3.0 (2.8–4.1) vs. 2.8 (1.6–3.0)	13.9 (11.6–16.0) vs. 10.6 (8.3–13.5)
				0 58				
				1 42				
				HBV+ 26				
				HCV+ 16			HR 0.718 (0.570–0.904)	HR 0.781 (0.611–0.998)
				Child-Pugh				
				A 99.6%				
				B 0.4%				
				BCLC stage			One-sided *p* = 0.0022	One-sided *p* = 0.0238
				B 20				
				C 80				
				Extrahepatic disease 70				
Qin [44]	Camrelizumab (q2w or Q3w)	II, randomized	220 (217)	PS	>1 prior systemic therapy 97	RR 14.7 (10.3–20.2)	PFS 2.1 (2.0–3.4)	13.8 (11.5–16.6)
				0 21				
				1 79				
				Child-Pugh		SD 29.5 (68.3–79.9)		
				A 98				
				B 2				
Pishvaian [45]	Cemiplimab	I, expansion (preliminary)	26 (26)	Not candidate for surgery	>1 prior systemic therapy 92	RR 19.2 SD 53.8	PFS 3.7 (2.3–9.1)	NR
				PS 0 23				
				PS 1 73				
Wainberg [46]	Durvalumab	I/II Various diagnosis; interim analysis HCC cohort (preliminary data)	40 (evaluable 39)	PS NR HBV+ 23 HCV+ 20 Child-Pugh A 100	Sorafenib 93	All Confirmed RR 10.3 (2.9–24.2)	NR	OS All, 13.2 (6.3–21.1) HBV+ 6.3 (1.4-NA) HCV+ 19.3 (9.5–23.0) Uninfected, 13.2 (4.7–24.2)
						DCR (SD ≥ 24 weeks) 33.3 (19.1–50.2)		
						HBV+, RR 0 (0–33.6)		
						DCR 11.1 (0.3–48.2)		
						HCV+, RR 25.0 (3.2–65.1)		
						DCR 62.5 (24.5–91.5)		
						Uninfected, RR 9.5 (1.2–30.4)		
						DCR 33.3 (14.6–57.0)		
Lee [47]	Avelumab	II	30	PS	Sorafenib 100	PR 10.0 DCR 73.3	TTP 4.4	14.2
				0 10				
				1 90				
				Child-Pugh				
				A 100				
				HBV+ 87				
				HCV+ 10				
**PD-1/PD-L1 inhibition in combination with antiangiogenic agents**								
Finn [51] (KEYNOTE-524)	Pembrolizumab *+* lenvatinib	Ib	100 (expansion phase)	PS	No prior therapy	RR 41 (31.1–51.3)	8.2 (95% CI 7.4–9.7)	22.0 (20.4–NE)
				0 62				
				1 38				
				HBV+ 19		CR 5		
				HCV+ 36				
				Child-Pugh				
				5 71		DCR 86 (95% CI 77.6–92.1)		
				6 27				
				7 2				
				BCLC		DOR 12.6 months (95% CI 6.2–18.7)		
				B 29				
				C 71				
Xu [52]	Camrelizumab *+* apatinib	I	18 (16 evaluable) (*+* gastric or esophageal junction cancer)	Advanced	Sorafenib 83	RR 50.0 (24.7–75.4) DCR 93.8 (69.8–99.8)	PFS 5.8 (2.6–not reached)	Not reached (4.0–not reached)
				PS				
				0 56				
				1 44				
				HBV+ 100				
				Child-Pugh				
				5 44				
				6 28				
				7 28				
				BCLC stage				
				B 6				
				C 94				
				Extrahepatic disease 89				
Xu [53] (RESCUE)	Camrelizumab *+* apatinib	II	190 (190) 70 (1. Line) 120 (2. line)	1. line	1. line 77/190 2. line 120/190	1. line RR 34.3 (95% CI 23.3–46.6) 2. line RR 22.5 (95% CI 15.4–31.0)	1. line 5.7 (95% CI 5.4–7.4) 2. line 5.5 (95% CI 3.7–5.6)	12-months survival rate 1. line 74.7 (95% CI 62.5–83.5) 2. line 68.2 (95% CI 59.0–75.7)
				PS				
				0 66				
				1 34				
				HBV+ 87				
				HCV+ 0				
				Child-Pugh				
				A 100				
				BCLC				
				B 17				
				C 83				
				2. line				
				PS				
				0 57				
				1 43				
				HBV+ 88				
				HCV+ 1				
				Child-Pugh				
				A 100				
				BCLC				
				B 18				
				C 82				
Jiao [55]	Penpulimab *+* anlotinib	Ib/II (preliminary data)	31 (25)	PS	No prior therapy	RR 24 DCR 84	6-months TTP	
				0 64				
				1 36			63% (95% CI 38–81)	
				HBV+ 61				
				HCV+ 7				
				BCLC			Median TTP not reached	
				B 23				
				C 77				
Lee [56]	Atezolizumab *+* bevacizumab	Ib (GO30140, arm A) (preliminary data)	104 (104)	PS	No prior systemic therapy	RR 36 (CR 12)	PFS 7.3 (95% CI 5.4–9.9)	17.1 (95% CI 13.8-NE)
				0 50				
				1 50				
				HBV+ 49		DCR 71		
				HCV+ 30				
				Child-Pugh				
				A5–6 94		DOR Not reached (95% 11.8-NE)		
				A 7 6				
				Extrahepatic disease or macrovascular invasion 88				
Lee [56]	Atezolizumab *+* bevacizumab vs. Atezolizumab	Ib (GO30140, arm F)	60 (60) vs. 59 (58)	PS	No prior systemic therapy	RR 20 vs. 17 DCR 67 vs. 49 DOR Not reached (NE) vs. Not reached (3.7–NE)	PFS 5.6 (3.6–7.4) vs. 3.4 (1.9–5.2)	Not reached (8.3–NE) vs. Not reached (8.2–NE)
				0 45 vs. 42				
				1 55 vs. 58				
				HBV+ 57 vs. 54				
				HCV+ 18 vs. 17			HR 0.55 (80% CI 0.40–0.74) *p* = 0.011	
				Child-Pugh				
				A 100 vs. 100				
				Extrahepatic disease or macrovascular invasion 78 vs. 85				
Finn [57,58]	Atezolizumab *+* bevacizumab vs. Sorafenib	III	336 vs. 165	PS	No prior systemic therapy	RR 29.8 (24.8, 35.0) vs. 11.3 (6.9, 17.3) CR 7.7 vs. 0.6 SD 44.2 vs. 43.4	PFS 6.8 (5.7–8.3) vs. 4.3 (4.0–5.6) HR 0.59 (0.47–0.76) *p* < 0.0001 6-months PFS 54.5% vs. 37.2%	12-month OS 67.2% vs. 54.6% median OS 19.2 months vs. 13.4 month (HR, 0.66; 95% CI, 0.52, 0.85; *p* = 0.0009)
				1 38 vs. 38				
				HBV+				
				49 vs. 46				
				HCV+				
				21 vs. 22				
				Child-Pugh				
				A5 72 vs. 73				
				A6 28 vs. 27				
				B7 0.3 vs. 0				
Kudo [59]	Avelumab *+* axitinib	I b (VEGF Liver 100) (preliminary data)	22	PS 0–1	No prior therapy	RR 13.6 (2.9–34.9)	NR	NA (immature)
				Child-Pugh A				
Bang [60]	Durvalumab *+* ramucirumab	Ib	28	PS	1 prior regimen 93	RR 11 DCR 61	PFS 4.4 (1.6–5.7)	10.7 (5.1–18.4)
				0 32				
				1 68				
				HBV+ 14				
				HCV+ 36				
				BCLC				
				B 21				
				C 79				
**PD-1/PD-L1 and CTLA-4 combination therapy**								
Yau [63] (CheckMate 040)	Nivolumab + ipilimumab	I/II	50 (50) 49 (49) 49 (49)	Advanced PS 0–1 Child-Pugh A 97 HBV+ 51 HCV+ 22 BCLC B 7 BCLC 91 Extrahepatic spread 82	Sorafenib 99	All patients RR 31(24–39) CR 5 DCR 49 RR 32 (20–47) 31 (18–45) 31 (18–45) DCR 54 43 49 DOR 17.5 months 4.6–30.5 *+)* 22.2 months (4.2–29.9*+)* 16.6 months (4.1–32.0*+)*	NR	22.8 (9.4–NE) 12.5 (7.6–16.4) 12.7 (7.4–33.0)
	Nivolumab 1 mg/kg + ipilimumab 3 mg/kg Q3W x 4→ nivolumab 240 mg Q2W							
	Nivolumab 3 mg/kg + ipilimumab 1 mg/kg Q3W x 4→ nivolumab 240 mg Q2W							
	Nivolumab 3 mg/kg Q2W+ ipilimumab 1 mg/kg Q6W							
Kelley [64]	Durvalumab *+* tremelimumab *	I/II (preliminary data)	40	Unresectable	No prior systemic therapy 30	Confirmed RR	NR	NR
				HBV+ 28		All 15		
				HCV+ 22		Uninfected 30		
				Child-Pugh		Infected 0		
				A 93		DCR (SD≥16 weeks) 57.5		
Kelley [65]	Durvalumab *+* tremelimumab 300 mg single dose Durvalumab *+* tremelimumab 75 mg Durvalumab Tremelumab	Randomized expansion cohort (preliminary data)	75	Advanced NR	No prior systemic therapy	22.7 (13.8–33.8)	NR	18.7 (10.8–not reached)
			84			9.5 (4.2–17.9)		11.3 (8.4–14.6)
			104			9.6 (4.7–17.0)		11.7 (8.5–16.9)
			69			7.2 (2.4–16.1)		17.1 (10.9–not reached)
Floudas [66]	Durvalumab *+* tremelimumab	II (preliminary data)	10 (*+* biliary cancer)	Advanced	At least one prior systemic therapy	PR 20	PFS 7.8 (2.6–10.6)	15.9 (7.1–16.3)
				NR		SD 40		
**PD-1/PD-L1 in combination with other drugs**								
Qin [68]	Camrelizumab *+* FOLFOX4 or GEMOX	II (preliminary)	34 (*+* biliary tract cancer)	Advanced HBV+ 79	No prior systemic therapy	RR 26.5	PFS 5.5	NR
						DCR 79.4		
						DOR not reached (3.3–11.5 + months)		
**PD-1/PD-L1 in combination with other treatment strategies**								
Lyu [69]	Nivolumab/pembrolizumab→nivolumab/pembrolizumab *+* subtotal thermal ablation	NR	50→33 (ablation)	Advanced	Prior sorafenib 100	PD-1 inhibitor RR 10 SD 42 *+* ablation RR 24	All patients 5 (95% CI 2.9–7.1)	All patients 16.9 (95% CI 7.7–26.1)
				PS				
				0 32				
				1 68				
				Child-Pugh				
				A 92				
				B 8				
				Extrahepatic metastases				
				74				
Tai [70]	Y90-radioembolization→Nivolumab	NR	40 (36 evaluable)	Advanced	Prior systemic therapy 14	RR 31.0 (95% CI 16.4–48.1)	4.6 (95% CI 2.3–8.4)	15.1 (95% CI 7.8–NE)
				Child-Pugh A				
				HBV+ 64		DCR 58.3		
				BCLC C 64				
**Preoperative; eligible for surgical resection**								
**CTLA-4 and PD-1/PD-L1 combination therapy**								
Kaseb [71]	Nivolumab vs. Nivolumab *+* ipilimumab	II, randomized (preliminary data)	13 vs.1421 (78%) proceed to surgery	Preoperative Eligible for surgical resection HBV+ 33 HCV+ 33	NR	pCR 24	NR	NR
						Major pathological response		
						16		
						among patients who proceed to surgery		
						pCR		
						19 among all randomized patients		

CA, cryoablation; CI, confidence interval; CR, complete response; DCR, disease control rate; DOR, duration of response; FOLFOX4, 5-flourouracil, leucovorin and oxaliplatin; GEMOX, gemcitabine and oxaliplatin; HBV, Hepatitis B virus; HCV, Hepatitis C virus; NA, not available; NE, not estimable; NR, not reported; OS, overall survival; pCR, pathological complete response; PFS, progression free survival; PR, partial response; PS, performance status; RFA, radiofrequency ablation; RR, response rate; SD, stable disease; TACE, chemoembolization; TTP, time to progression.

**Table 3 jcm-10-02662-t003:** Treatment-related overall adverse events and hepatotoxicity of checkpoints inhibitors in HCC.

	Treatment	Number of Patients Included	Adverse Event	Adverse Event, Grade 3–4	Increased AST, Any Grade	Increased AST, Grade 3–4	Increased ALT, Any Grade	Increased ALT, Grade 3–4	Hepatitis, Any Grade	Hepatitis, Grade 3–4	Discontinuation Due to Toxicity; Grade 5 Adverse Event (%)
**Advanced, not amenable for resection or ablation**											
**CTLA-4 inhibition**											
Sangro ^1^ [33]	Tremelimumab	21	-	-	70	45	55	25	0	0	NR
											Grade 5 0
Duffy ^2^ [36]	Tremelimumab +	32	-	-	34 (≥grade 2)	22	19 (≥grade 2)	9	0	0	13
	RFA/CA/TACE										Grade 5 0
**PD-1/PD-L1 inhibition; monotherapy**											
El-Khouiry [37] (CheckMate 040)	Nivolumab	48	83	25	21	10	15	6	0	0	2
											Grade 5 0
	Nivolumab	214	74	19	7	4	8	2	0	0	4
											Grade 5 0
Kudo [38]	Nivolumab	49	51	-	-	-	-		8	-	4
(CheckMate 040; Child-Pugh B cohort)											Grade 5 NR
Yau [39] (CheckMate 459)	Nivolumab	371 vs. 372	-	22 vs. 49	-	-	-	-	-	-	4 vs. 8
	Sorafenib										Grade 5 NR
Zhu [40] (Keynote-224)	Pembrolizumab	104	73	24	14	7	9	5	3	-	5
											Grade 5 1
Feun ^1^ [42]	Pembrolizumab	29 (evaluable 28)	76	10	28	17	34	7			6
											Grade 5 NR
Finn ^1^ [43] (Keynote-240)	Pembrolizumab vs. Best supportive care	278 vs. 135	96 vs. 90	52 vs. 46 19 vs. 5 (TRAE)	23 vs. 16	13 vs. 8	18 vs. 10	6 vs. 3	3 vs. 0	1 vs. 0	17 vs. 9 Grade 5 2.5 vs. 3.0
Qin [44]	Camrelizumab (q2w or Q3w)	220 (217)	-	-	26	5	23	2	2.3 (hepatic function abnormal)	2.3	4
			(RCCEP 67)	(RCCEP 0)							Grade 5 0.9 (hepatic failure 0.5)
Pishvaian [45]	Cemiplimab	26	-	-	23	8	-	-	-	8	NR
											Grade 5 7.7
Wainberg [46]	Durvalumab	40	80	20	23	8	-	5	-	-	18
											Grade 5 0
Lee [47]	Avelumab	30	77	23	AST/ALT	AST/ALT	-	-	-	-	7
					23	13					Grade 5 0
**PD-1/PD-L1 inhibition in combination with antiangiogenic agents**											
Finn [51]	Pembrolizumab *+* lenvatinib	100	99	67	30	14	19	6	-	-	18
				(hypertension 17)							Grade 5 3
Xu [52]	Camrelizumab *+* apatinib	33 (*+* gastric or esophageal junction cancer; dose expansion phase)	-	-	52	15	39	9	-	-	9
											Grade 5 0
Xu [53]	Camrelizumab *+* apatinib	190 (190)	99.5	774	63	20	53	7	3 (hepatotoxicity)	3	12
			(hypertension 73; RCCEP 30)	(hypertension 34; RCCEP 0.5)							Grade 5 1.1
Jiao [55]	Penpulimab *+* anlotinib	40 (36)	94	10	36	-	29	-	-	-	7
											Grade 5 0
Lee [56]	Atezolizumab *+* bevacizumab	104 (104)	88	53 (AE) (hypertension 14)	15 (AEsi)	5 (AEsi)	12 (AEsi)	3 (AEsi)	1 (AEsi)	1 (AEsi)	10
											Grade 5 3
Lee [56]	Atezolizumab *+* bevacizumab vs. Atezolizumab	60 (60)	68 vs. 41	20 vs. 5	5 vs. 14 (AEsi)	3 vs. 3 (AEsi)-	5 vs. 9 (AEsi)-	0 vs. 0	0 vs. 0	0 vs. 0	2 vs. 1
		59 (58)		(hypertension 5 vs. 1)							Grade 5 0
Finn [57,58]	Atezolizumab *+* bevacizumab vs. Sorafenib	336 vs. 165	98 vs. 98	56.51 vs. 55.1	-	-	-	-	0 vs. 1.3 (hepatic failure)	0 vs. 1.3 (hepatic failure)	16 vs. 10
				(hypertension 15.2 vs. 12.2)							Grade 5 4.6 vs. 5.8
Kudo [59]	Avelumab *+* axitinib	22	-	Hypertension 50	-	-	-	-	-	0	0; no grade ≥3 immune-related AE
				Hand-foot syndrome 23							Grade 5 0
Bang ^3^ [60]	Durvalumab + ramucirumab	28	86	43 Hypertension 18	25	18	7	4	-	-	18 Grade 5 7
**PD-1/PD-L1 and CTLA-4 combination therapy**											
Yau [63] (CheckMate 040)	Nivolumab *+* ipilimumab (3 dosing regimens)	50 (50) 49 (49) 49 (49)	94 71 79	53 29 31	20 20 13	16 14 8	16 14 8	8 6 0	20 12 6	20 10 6	18
											6
											2
											Grade 5
											1
											0
											0
Kelley [64]	Durvalumab *+* tremelimumab	40	60	20	15	10	18	-	-	-	8
											Grade 5 0
Kelley [65]	Durvalumab *+* tremelimumab 300 mg single dose Durvalumab *+* tremelimumab 75 mg Durvalumab Tremelumab	75 84 104 69	-	35 24 18 42	-	-	-	-	-	-	11
											6
											8
											12
											Grade 5
											0
											1/84
											3/104
											0
Floudas [66]	Durvalumab *+* tremelimumab	10 (*+* biliary cancer)	-	-	-	-	-	-	-	-	-
**PD-1/PD-L1 in combination with other drugs**											
Qin [68]	Camrelizumab *+* FOLFOX4 or GEMOX	34 (*+* biliary tract cancer)	-	85	-	-	-	-	-	-	0
											Grade 5 NR
**PD-1/PD-L1 in combination with other treatment strategies**											
Lyu [69]	Nivolumab/pembrolizumab →nivolumab/pembrolizumab *+* subtotal thermal ablation	50	82	-	20 (*+* ALAT)	0 (*+* ALAT)	-	-	0	0	8
											Grade 5 1/50
Tai [70]	Y90-radioembolization→Nivolumab	40 (36)	-	11	-	-	-	-	-	-	-
**Preoperative; eligible for surgical resection**											
**CTLA-4 and PD-1/PD-L1 combination therapy**											
Kaseb [71]	Nivolumab vs. Nivolumab *+* ipilimumab	13 vs. 14	-	5 vs. 24	-	-	-	-	-	-	-
		21 (78%) proceed to surgery									Grade 5 0

AE, adverse event; RCCEP, reactive cutaneous capillary endothelial proliferation; TRAE, treatment-related adverse event. ^1.^ All AEs; ^2.^ All AEs ≥ grade 2; ^3.^ Treatment emergent AEs.

**Table 4 jcm-10-02662-t004:** Correlation between response and PD-L1 expression.

	Treatment	Number of Patients in CPI Arm	Number of Tissue Samples Available	Result
El-Khouiry [37] (CheckMate 040)	Nivolumab	48	44	PD-L1 assessed by membrane expression on tumor cells:
				Positive (≥1%) (n = 11) RR 27%; negative (<1%) (n = 33) RR 12%; *p*-value NR
	Nivolumab	214	174	PD-L1 assessed by membrane expression on tumor cells:
				Positive (≥1%) (n = 34) RR 26%; negative (<1%) (n = 140) RR 19%; *p*-value NR
Sangro [105] (CheckMate 040)	Nivolumab	262	PD-L1 195	PD-L1 assessed by membrane expression on tumor cells:
				Positive (≥1%) (n = 36) RR 28% (16–44); negative (<1%) (n = 159) RR 16% (11–22); *p*-value NR
				Positive (≥1%) (n = 36) median OS 28.1 months (95% CI 18.2-NA); negative (<1%) (n = 159); median OS 16.6 months (95% CI 14.2–20.2) (*p* = 0.032)
Yau [39] (CheckMate 459)	Nivolumab	371	366	PD-L1 assessed by tumor positive score:
	Sorafenib			Positive (≥1%) (n = 71): RR 28%; negative (<1%) (n = 295) RR 12%; (*p*-value NR)
Zhu [40] (Keynote-224), Kudo [41]	Pembrolizumab	104	52	PD-L1 assessed by combined positive score (CPS) (a measure of PD-L1 positive immune and tumor cell number):
				Positive (≥1%) (n = 22): RR 32%; negative (<1%) (n = 30) RR 20%; *p* = 0.021
				PD-L1 assessed by tumor positive score (TPS):
				Positive (≥1%) (n = 7): RR 43%; negative (<1%) (n = 45) RR 22%; *p* = 0.088
Feun [42]	Pembrolizumab	29	10	Method for PD-L1 assessment NR:
				Positive (level NR) (n = 4) RR 25%; negative (level NR) (n = 6) RR 33%
Qin [44]	Camrelizumab (q2w or Q3w)	220	30	PD-L1 assessed by tumor proportion score:
				Positive (≥1%) (n = 11) RR 36%; negative (<1%) (n = 19) RR 11% (*p*-value NR)
Lee [47]	Avelumab	30	27	Four antibodies were investigated with different evaluation methods used for each clone:
				Expression of PD-L1 was not associated with response
Xu [52]	Camrelizumab *+* apatinib	43 (*+* gastric or esophageal junction cancer)	18 (type of cancer NR)39	PD-L1 assessed on circulating tumor cells (CTC):
				High (≥20%) RR 48%; Low (<20%) RR 0%; *p* = 0.002; PFS significantly longer in patients with high expression compared to low (HR 0.28; *p* = 0.0002; OS not significantly different (HR 0.40; *p* = 0.601)
Xu [53]	Camrelizumab *+* apatinib	190	54	PD-L1 assessed by tumor proportion score:
				RR and PFS similar between positive (≥1%) and negative (<1%) patients
Lee [56]	Atezolizumab *+* bevacizumab	104	86	PD-L1 assessed on tumor cells and tumor-infiltrating immune cells:
				Positive ≥ 1% (n = 61) RR 41%; ≥ 5% (n = 37) RR 46%; ≥ 10% (n = 30) RR 50%
				Negative < 1% (n = 25) RR 28%; < 5% (n = 49) RR 31%; < 10%(n = 56) RR 30%
				(*p*-values NR)
Lee [56]	Atezolizumab *+* bevacizumab Atezolizumab	60 59	95	PD-L1 assessed on tumor cells and tumor-infiltrating immune cells:
				Positive ≥ 1% (n = 62) PFS 5.6 months; ≥ 5% (n = 24) PFS 4.1 months; ≥ 10% (n = 11) PFS 3.7 months
				Negative < 1% (n = 33) PFS 5.7 months; < 5% (n = 71) PFS 5.7 months; < 10% (n = 84) PFS 5.7 months (*p*-value NR)
				Positive ≥ 1% (n = 62) PFS 2.1 months; ≥ 5% (n = 24) PFS 1.9 months; ≥ 10% (n = 11) PFS 2.7 months
				Negative < 1% (n = 33) PFS 4.0 months; < 5% (n = 71) PFS 3.7; < 10% (n = 84) PFS 5.4 months
				(*p*-values NR)
Bang [60]	Durvalumab *+* ramucirumab	28	26	PD-L1 assessed on tumor cells:
				High (≥25%) (n = 11) RR 18% and SD 73%; low (<25%) (n = 15) RR 0 and SD 47% (*p*-values NR)
Yau [63]	Nivolumab Ipilimumab	50 49 49	49 48 48	Method for PD-L1 assessment NR:
				Positive (≥1%) (n = 10) RR 30%, OS 18.8 months; negative (<1%) (n = 39) RR 31%, OS 22.2 months
				Positive (≥1%) (n = 10) RR 30%, OS 10.2 months; negative (<1%) (n = 38) RR 32%, OS 12.5 months
				Positive (≥1%) (n = 8) RR 50%, OS NE; negative (<1%) (n = 40) RR 28% OS 10.4 months

CTC, circulating tumor cells; NR, not reported, HR, hazard ratio; OS, overall survival; NA, not available; PFS, progression-free survival; RR, response rate; SD, stable disease.

## Figures and Tables

**Figure 1 jcm-10-02662-f001:**
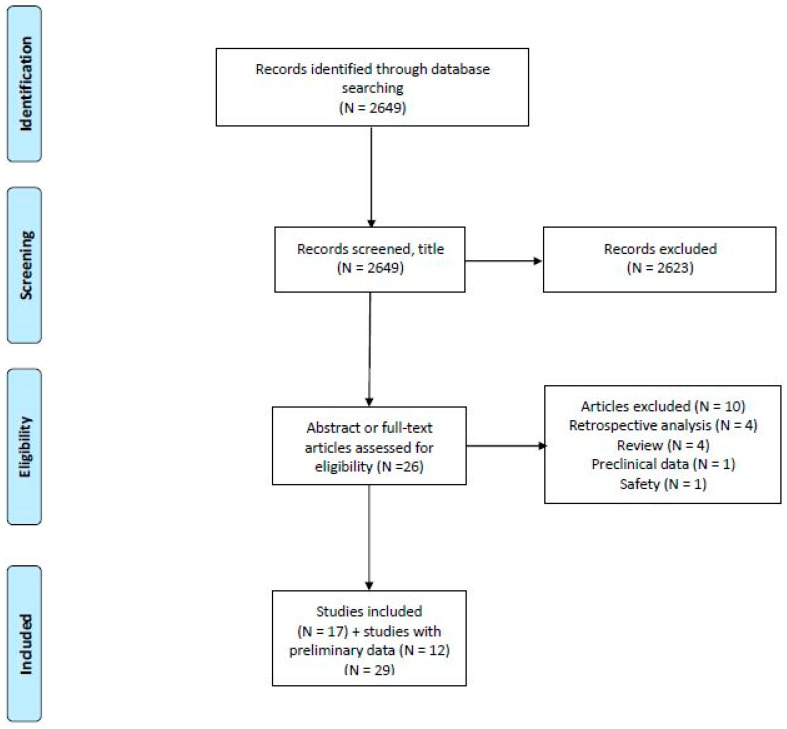
PRISMA flow diagram.

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
