# Peer review of "Clinical Trials of Immune Checkpoint Inhibitors in Hepatocellular Carcinoma"

_jcm, 2021, doi:10.3390/jcm10122662_

Round 1

Reviewer 1 Report

I read with interest the comprehensive review by Dyhl-Polk et al. discussing published and ongoing results on immune checkpoint inhibitors (CPIs) in the setting of hepatocellular carcinoma.

The authors reported all the major results derived from several CPIs under evaluation in different clinical trials. In addition they included in their revision preliminary data from ongoing clinical trials. Overall, is a well-written manuscript and I do not have particular comments.

Minor revision

  • I suggest to summerize the body of the abstract (particularly the results section) focusing on the most important messages in the field)
  • In the flow diagram, the authors included ongoing studies with preliminary data at the bottom of the diagram, with the number of the studies considered in the review. I think that these studies should be included with all the studies considered in the first block (2649 + 12, 2661) and decribed later in the flow chart.

Author Response

Reviewer 1:

I read with interest the comprehensive review by Dyhl-Polk et al. discussing published and ongoing results on immune checkpoint inhibitors (CPIs) in the setting of hepatocellular carcinoma.

The authors reported all the major results derived from several CPIs under evaluation in different clinical trials. In addition they included in their revision preliminary data from ongoing clinical trials. Overall, is a well-written manuscript and I do not have particular comments.

Response: We thank the reviewer for this positive evaluation of the current manuscript.

Minor revision

I suggest to summerize the body of the abstract (particularly the results section) focusing on the most important messages in the field)

Response: The abstract has been changed as suggested.

In the flow diagram, the authors included ongoing studies with preliminary data at the bottom of the diagram, with the number of the studies considered in the review. I think that these studies should be included with all the studies considered in the first block (2649 + 12, 2661) and decribed later in the flow chart.

Response: Those studies are already included in the first block. The first block includes all records identified through database searching – both conference abstracts (Embase) and full-text articles (Pubmed + Embase).

Reviewer 2 Report

Dyhl-Polk et al provide a structured review of clinical trials of immune checkpoint inhibitors in hepatocellular carcinoma. The manuscript gives an actual overview of different studies, which are sorted, summarized, and compared according to their setting of immune checkpoint administration +/- further therapeutic modalities. The introduction gives an overview, potentially predictive biomarkers are thoroughly discussed, ongoing trials are provided in the supplemental table. I have only minor issues.

Minor issues:

  • Lines 108-110: The quality of the correlation quoted in [24] should be named, too. It is important not only to state that there is a correlation but also whether it is negative or positive. In this case: Reduced lymphocytic infiltration was associated with recurrence after transplantation.
  • Lines 110-111: See above, please name whether the data of quote [25] showed a positive or negative correlation with survival.
  • Line 163: Please check position of quote [34]. Please also check / correct in the references part!
  • Lines 169, 170: Quote [33] would be more suitable (primary source)
  • Line 172: Please check position of quote [34].
  • Line 568: “camreziluma” -> camrezilumab
  • Line 603: “patient” -> patients
  • Line 740: “an” -> a
  • Figure 1: Please correct under “Eligibility”: “26” -> N=26; “Review (N04)” -> Review (N=4)
  • Table 1: Please harmonize fonts to improve readability.
  • Table 2: Please improve readability. E.g. first column could be smaller or landscape format could be used for the pages (if possible).
  • Table 3: Please improve readability analogue to Table 2.

Author Response

Please see our response in the attached word-file
